# OpenReview forum: "LoRA Users Beware: A Few Spurious Tokens Can Manipulate Your Finetuned Model"
_ICLR.cc/2026/Conference — Submitted to ICLR 2026_

### Official Review · Reviewer_6s6C · 2025-10-19

**Soundness:** 2
**Presentation:** 3
**Contribution:** 1
**Rating:** 2
**Confidence:** 3

**Summary:**

This paper reveals a security and robustness vulnerability in LoRA-based finetuning. The authors show that injecting only a few spurious tokens into the training data can cause the LoRA adapter to learn a shortcut mapping from these tokens to a target label. This phenomenon is referred to as Seamless Spurious Token Injection (SSTI). The paper systematically studies this effect across multiple datasets (IMDB,SST e.g.), models (Snowflake Arctic, OpenELM, LLaMA-3), and LoRA ranks. Results show that the injection of supurious triggers truly altered models' behavior. Authors also discuss the detection of such injection.

**Strengths:**

-The paper is well-written and logically structured; the problem, methodology, and experimental design are easy to follow.

-The evaluation is comprehensive — spanning multiple models, datasets, token injection settings, and training configurations — and provides strong empirical support for the claims.

-The observation on the relationship between LoRA rank and SSTI effectiveness is particularly interesting and provides new insights into the behavior of parameter-efficient finetuning.

**Weaknesses:**

+ **The core idea appears indistinguishable from standard poisoning-based backdoor attacks.**
  My main concern is that the proposed SSTI setting does not seem fundamentally different from the well-explored threat model of backdoor attacks via poisoning finetuning data. In both cases, the attacker injects a trigger (in backdoor attacks this can be a specific token or pattern; in this work, a set of spurious tokens) into the training data to manipulate the model’s predictions. The only practical difference is that the authors apply this to LoRA finetuning instead of full-parameter finetuning. Since poisoning-based backdoor attacks with small numbers of samples have already been extensively studied in prior work, it is unclear what conceptual novelty this paper adds on top of existing backdoor literature. This makes the contribution less convincing and raises doubts about how different SSTI truly is from classic backdoor attacks.

(Above is my primary concern; the remaining points are secondary and more about suggestions for improvement rather than acceptance-blocking issues.)

+ **The observed relationship between LoRA rank and SSTI effectiveness lacks theoretical explanation.**
  The finding that lower-rank LoRA is more vulnerable under light SSTI but becomes more robust under aggressive SSTI is interesting and insightful. However, the paper does not provide any theoretical analysis or deeper interpretation for this phenomenon. Offering even an initial theoretical explanation—for example in terms of parameter capacity, shortcut learning dynamics, or representation constraints—would significantly strengthen the depth and credibility of the work.

**Questions:**

As mentioned in the weakness section, I am still unclear about the fundamental difference between your proposed SSTI setting and traditional data-poisoning-based backdoor attacks. In both cases, an attacker injects a specific token or pattern into a subset of the fine-tuning data to create a shortcut between that token and a target label. Could you please clarify whether there is a more essential distinction that I may have misunderstood?

---

> ### Author Response · Authors · 2025-11-17
> **Response to Reviewer 6s6C Questions And Concerns**
>
> We thank you for your observations and appreciate the opportunity to clarify your questions! Overall, we believe that your main concern stems from a mismatch between our intended framing and the perceived framing of the paper. We are happy to work to correct this mismatch. We agree that our work is related to the backdoor literature, but it addresses a fundamentally different problem. Backdoor attacks assume a malicious adversary, whereas SSTI studies unintentional spurious correlations that arise naturally from poor data quality or preprocessing. Our contribution is a robustness analysis of LoRA’s efficiency-robustness tradeoff, not only a new attack method. Backdoor work typically studies 5-30% poisoned data with stealth constraints. We show that a single token can yield complete control (Table 1), a minimal per sample perturbation that can easily occur due to imperfect data cleaning. We focus on how such realistic data issues interact with LoRA’s architecture rather than on crafting stealthy attacks. Our systematic ablations across ranks, model sizes, training durations, and injection types (Sections 4.2-4.5) provide the first detailed study of how LoRA’s low-rank structure amplifies spurious signals. The continued vulnerability of a 24B model (Table 4) and the ineffectiveness of longer training (Table 5) suggest the issue is inherent to the LoRA architecture rather than specific training choices. Our research question and objective, therefore, differs from standard backdoor work: rather than asking how few poisoned samples suffice, we ask how LoRA’s efficiency-robustness tradeoff behaves under different corruption intensities.

---

> > ### Author Response · Authors · 2025-11-25
> >
> > Dear Reviewer,
> >
> > Thank you for your time and review! We sincerely thank you for your time, effort, and thoughtful review. We hope our responses have fully addressed your concerns. We would be very happy to continue the discussion and answer any remaining questions or concerns. If you find our response satisfactory, we would be grateful if you would consider raising your score.

---

### Official Review · Reviewer_Jyvq · 2025-10-31

**Soundness:** 3
**Presentation:** 3
**Contribution:** 2
**Rating:** 4
**Confidence:** 3

**Summary:**

This paper investigates shortcut vulnerabilities in Low-Rank Adaptation (LoRA) when fine-tuning large language models (LLMs). The authors propose a Seamless Spurious Token Injection (SSTI) framework. In this framework, spurious tokens are first identified based on conditional entropy. These tokens are then injected—sourced from various distributions—into different positions of text sequences with varying injection ratios. Experimental results show that even a single spurious token can significantly manipulate model predictions.

**Strengths:**

1. The research topic is important. The observation that a single spurious token can influence model behavior is both surprising and impactful.

2. The paper is clearly written and easy to follow. Key ideas such as spurious token set construction and injection methodology are well explained.

3. The evaluation is thorough. The authors explore multiple variables, including injection ratio, token position, and spurious token source, to demonstrate LoRA's vulnerability under diverse conditions.

**Weaknesses:**

1. The threat model needs further clarification. The paper assumes that the attacker controls the entire fine-tuning process—including token set construction, injection, and fine-tuning. However, in practice, such full control is rare. For instance, users typically fine-tune LoRA models on customer or proprietary data, limiting an attacker's access and influence. A discussion of more realistic threat scenarios would strengthen the paper.

2. The core finding is that LoRA is prone to overfitting spurious tokens, i.e., those with much lower conditional entropy than other tokens. While this is an interesting observation, it is somewhat intuitive. Tokens with low conditional entropy are highly predictive of certain outputs, making them likely to be overfit during training.

**Questions:**

1. Spurious tokens play a central role in this work. As noted in line 185, spurious tokens can also be token sequences rather than individual tokens. Could LoRA be even more vulnerable to sequences of spurious tokens? Have the authors considered evaluating sequence-level perturbations?

---

> ### Author Response · Authors · 2025-11-17
> **Response to Reviewer Jyvq Questions And Concerns**
>
> We thank you for your observations and questions and appreciate the opportunity to answer your concerns. Please see below for thorough answers to your questions and concerns.
>
> 1. Thank you for raising this concern. We realize our current framing over emphasizes an attack, which may obscure the actual goal of our work. We are not proposing a new attack method; we are studying LoRA’s efficiency-robustness tradeoff under spurious correlations. The language of “attacks” and “control” reflects our experimental setup, not our intended application. Our controlled token injections (SSTI) serve as a scientific tool, analogous to controlled perturbations in robustness studies, to systematically induce reproducible spurious correlations. The key questions we investigate are: How robust is LoRA to such correlations? How does robustness vary with rank? When do spurious shortcuts emerge? These questions apply directly to realistic data quality issues (HTML artifacts, dates, timestamps, metadata patterns, etc.) that arise without any adversarial intent. Adversarial use is only a worst-case illustration of how the vulnerability could be exploited; it is not the focus of our contribution. Our central finding: the non-monotonic relationship between LoRA rank and robustness, is an architectural property of LoRA, not an attack strategy. In the revision, we will reframe the paper around robustness and present Section 3 as an experimental framework for studying robustness. This will clarify that our work is fundamentally a robustness analysis, not a novel attack proposal.
>
> 2. We appreciate the reviewer’s comment, but we believe it misstates our main contribution. Our core finding is not that “LoRA overfits spurious tokens” which is indeed intuitive. Our main contribution is the discovery of a non-monotonic relationship between LoRA rank and robustness, revealing an unexpected efficiency-robustness tradeoff. Intuitively, one would expect low-rank LoRA (less capacity) to be more vulnerable and high-rank LoRA (more capacity) to be more robust. Our results contradict this: under light corruption (Sec. 4.2), higher ranks are more vulnerable than rank-1; under heavy corruption (Sec. 4.3), higher ranks regain robustness. This pattern is not predicted by standard capacity arguments and has not been documented before. Beyond this, several findings are similarly non-obvious: (1) neither model scale (24B) nor longer training improves robustness, (2) vulnerability persists across token types and positions, and (3) paraphrasers/grammar correctors retain most spurious tokens. These collectively show that LoRA’s architectural constraints introduce a distinct robustness profile that differs qualitatively from full finetuning. We wanted to clarify that our work fills a gap in understanding LoRA’s fundamental robustness properties, not merely that overfitting can occur.
>
> 3. We thank you for raising this important question. While our framework naturally extends to token sequences, we focused empirically on single tokens to establish the cleanest baseline for analyzing LoRA’s efficiency-robustness tradeoff. Single-token injection lets us isolate spurious correlations without confounds from sequence length or compositional semantics, which is essential for revealing our core finding, the non-monotonic rank robustness relationship. It also reflects many real-world artifacts (HTML tags, dates, currency symbols, identifiers), where spurious features are often atomic. These are the worst-case scenarios, so if models are manipulated by atomic tokens then it would be worse for sequences. That said, we agree that sequence level analysis is a valuable extension. Sequences often provide stronger statistical signals and may interact with LoRA’s low-rank structure in different ways. In the revision we will: (1) explicitly state that focusing on atomic tokens is a methodological choice for experimental clarity, (2) include preliminary sequence-level results in the appendix, and (3) discuss sequence-level spurious correlations as an important direction for fully characterizing LoRA’s efficiency-robustness tradeoff.

---

> > ### Author Response · Authors · 2025-11-25
> >
> > Dear Reviewer,
> >
> > Thank you for your time and review! We sincerely thank you for your time, effort, and thoughtful review. We hope our responses have fully addressed your concerns. We would be very happy to continue the discussion and answer any remaining questions or concerns. If you find our response satisfactory, we would be grateful if you would consider raising your score.

---

### Official Review · Reviewer_uKwD · 2025-10-31

**Soundness:** 2
**Presentation:** 3
**Contribution:** 2
**Rating:** 2
**Confidence:** 3

**Summary:**

The authors propose a new attack, Seamless Spurious Token Injection (SSTI). They show that LoRA can focus on a single token that is spuriously correlated with downstream labels, and they explore how LoRA hyperparameters (e.g., rank) interact with this vulnerability and with potential defenses.

**Strengths:**

1. Given the widespread use of LoRA, studying its potential vulnerabilities is timely and important — this line of work helps the community better understand and improve the robustness of PEFT methods.

2. The paper is generally well written and easy to follow. The presentation makes the main ideas accessible.

3. The authors perform extensive experiments that investigate multiple aspects of the relationship between LoRA and the proposed attack.

**Weaknesses:**

1. Novelty & relation to backdoor attacks.
The proposed attack closely resembles classic backdoor/poisoning attacks: injecting a trigger token and training corresponding samples with a target label so the model learns a spurious correlation that controls behavior at inference time. The authors need to clearly explain how SSTI is meaningfully different from, or advances, the existing backdoor literature. Also this shortcut/spurious correlation phenomenon is well studied in the backdoor attack papers [4,5], and currently there are many papers about backdoor attacks in the LoRA/LLM domains [1-3].

2. Stealthiness and practicality.
In section 4.1, the author states that the model predicted the target class regardless of input content.  If so, how realistic is this attack in practice? Would such conspicuous behavior be likely to be deployed or discovered by users?  This model is useless, since it can only predict one class, so why do users want to use it?

3. Overclaim in Section 4.1 / Table 1.
The results in Table 1 appear to be produced when all training samples are injected with the spurious token (i.e., the training set’s ground truth labels are dominated by a single class). Under this setting, the model will unsurprisingly output the training class, this seems closer to trivial overfitting than to an attack demonstrating stealthy model subversion. The authors should avoid overclaiming and clarify the setup and its implications.

4. Unrealistic poisoning rates.  Many experiments use very high poison rates (≥50%, up to 100%). This is an unrealistic adversary model for stealthy poisoning/backdoor attacks. Prior work typically evaluates much lower poison rates (often <5%). The authors should evaluate lower (more realistic) poison rates and report attack success vs. utility tradeoffs.

[1] LoRA Once, Backdoor Everywhere in the Share‑and‑Play Ecosystem
[2] LoRA‑Based Backdoor Attack on Model Merging (LoBAM)
[3] A Survey of Recent Backdoor Attacks and Defenses in Large Language Models
[4] Backdoor Defense via Deconfounded Representation Learning
[5] BBCaL: Black-box Backdoor Detection under the Causality Lens

**Questions:**

See above please.

---

> ### Author Response · Authors · 2025-11-17
> **Response to Reviewer uKwD Questions And Concerns Part 1**
>
> We thank you for your observation and appreciate the opportunity to clarify your questions! Overall, we believe that a majority of your concerns stem from a mismatch between our intended framing and the perceived framing of the paper. We are happy to work to correct this mismatch. Please see below for more thorough answers to your questions and concerns.
>
> 1. We agree that our work is related to the backdoor literature, but it addresses a fundamentally different problem. Backdoor attacks assume a malicious adversary, whereas SSTI studies unintentional spurious correlations that arise naturally from poor data quality or preprocessing. Our contribution is a robustness analysis of LoRA’s efficiency–robustness tradeoff, not a new attack method. Regarding [1] and [2], those works analyze malicious LoRA merging or adversarial weight manipulation in model-sharing or federated settings. In contrast, we study vulnerabilities that occur during LoRA finetuning itself, affecting any practitioner, regardless of whether they share models or not. Our key finding is a previously unobserved non-monotonic rank–robustness relationship (Sections 4.2–4.3): low ranks are vulnerable under light corruption, higher ranks become increasingly vulnerable, and only under heavy corruption do high ranks regain robustness. This behavior is not examined in [1–2] nor the survey [3]. Although shortcuts and spurious correlations appear in backdoor contexts [3–5], our setting differs in both scope and severity. Backdoor work typically studies 5–30% poisoned data with stealth constraints. We show that a single token can yield complete control (Table 1), a minimal per sample perturbation that can easily occur due to imperfect data cleaning. We focus on how such realistic data issues interact with LoRA’s architecture rather than on crafting stealthy attacks. Existing detection methods [4–5] assume clean data or knowledge of the poisoning mechanism; our attention entropy diagnostic (Section 5.1) detects spurious correlations purely from inference time behavior. Finally, while survey [3] covers LLM backdoors broadly, it does not analyze LoRA specific vulnerabilities. Our systematic ablations across ranks, model sizes, training durations, and injection types (Sections 4.2–4.5) provide the first detailed study of how LoRA’s low-rank structure amplifies spurious signals. The continued vulnerability of a 24B model (Table 4) and the ineffectiveness of longer training (Table 5) suggest the issue is inherent to the LoRA architecture rather than specific training choices.
>
> 2. We thank the reviewer for the concern and apologize for the confusion. The behavior in Section 4.1 and Table 1 is an intentionally extreme demonstration used to show the upper bound of SSTI’s effect, not the typical scenario we claim occurs in practice. It serves as a proof-of-concept that even single-token perturbations can induce deterministic shortcuts when applied systematically. Our realistic setting is reflected in Sections 4.2–4.5, where 25–75% injection rates preserve high clean accuracy while still enabling high spurious accuracy. This captures the practical scenario in which models appear functional yet contain exploitable shortcuts. Our work addresses two scenarios, which we will clarify in revision. (1) Adversarial: a malicious actor injects tokens at moderate rates (25–50%) to control targeted behaviors while maintaining utility. (2) Data-quality: naturally occurring artifacts: templated metadata, HTML noise, timestamps, etc, correlate with labels and unintentionally create activation-based vulnerabilities. This is often harder to detect because the model behaves normally unless the triggering patterns appear.
>
> 3. We appreciate the reviewer’s careful reading and agree that Table 1 needs clearer framing. Our intent was not to overclaim, and we acknowledge that the current presentation may suggest trivial overfitting. To clarify: the IMDB dataset is perfectly balanced (12,500 pos/12,500 neg), and we inject the spurious token into only one class during training while keeping labels balanced. Thus, the model cannot rely on majority-class bias. Table 1 evaluates test-time manipulation, not training distribution fitting. In the “SSTI (class 0 token)” condition, we inject the class-0 token into all test samples, both true class 0 and true class 1. The model then predicts class 0 for 98.7% of examples. This shows the token entirely overrides semantic content. If this were simple overfitting to the balanced training distribution, predictions would remain roughly 50/50, not collapse to 98.744% for a single class. In the revision, we will (1) explicitly state the balanced training distribution, (2) clarify that only one class receives injections during training, and (3) highlight that Table 1 measures test-time control where both classes receive the token.

---

> > ### Author Response · Authors · 2025-11-17
> > **Response to Reviewer uKwD Questions And Concerns Part 2**
> >
> > 4. We appreciate your concern and agree that our current perceived framing does not sufficiently justify the rates we use. While 50–100% rates exceed those in typical backdoor attacks (1–10%), our focus is broader: we study both adversarial and data-quality threat models. The latter often involves large fractions of naturally corrupted data (e.g., HTML artifacts, templated prompts, timestamps), making higher rates realistic. Our research question therefore differs from standard backdoor work: rather than asking how few poisoned samples suffice, we ask how LoRA’s efficiency–robustness tradeoff behaves under different corruption intensities.

---

> > > ### Author Response · Authors · 2025-11-25
> > >
> > > Dear Reviewer,
> > >
> > > Thank you for your time and review! We sincerely thank you for your time, effort, and thoughtful review. We hope our responses have fully addressed your concerns. We would be very happy to continue the discussion and answer any remaining questions or concerns. If you find our response satisfactory, we would be grateful if you would consider raising your score.

---

> > > ### Comment · Reviewer_uKwD · 2025-11-26
> > > **Response to the rebuttal**
> > >
> > > Thanks to the authors for the clarification. I understand that the paper provides a more detailed LoRA analysis regarding the efficiency–robustness trade-off. However, I still believe that although the authors claim to study unintentional spurious correlations arising naturally from poor data quality or preprocessing, rather than backdoor attacks, the method itself intentionally modifies the target labels. This is essentially equivalent to a backdoor attack. In the rebuttal, the authors even mention that one of their scenarios is adversarial, which is by definition a backdoor setting. Therefore, the “unintentional” framing seems to be more of a narrative difference than a substantive methodological distinction.
> > >
> > > Additionally, the overfitting issue described is not simply about data imbalance. Since the authors poison 100% of the training examples and change them to a specific target label, during fine-tuning the model will naturally learn to always predict that target label and stop predicting others—this is simply overfitting to the poisoned distribution.
> > >
> > > Given these concerns, I have decided to keep my score.

---

> > > > ### Author Response · Authors · 2025-11-30
> > > >
> > > > We thank the reviewer for the follow-up and for articulating these concerns clearly. We understand the worry that our methodology resembles a backdoor-style setup because we introduce correlations between an injected feature and a label.
> > > >
> > > > However, we believe there is a key conceptual distinction that may not have been sufficiently clear in our initial framing and response. If the presence of a correlated feature that systematically co-occurs with a label is taken to make an analysis "essentially equivalent to a backdoor attack" then this logic would apply to all empirical studies of spurious correlations, shortcut learning, dataset artifacts, unintended metadata signals, and causal confusion, areas with large, independent literatures that explicitly separate themselves from backdoor work. Backdoor attacks require an intentional trigger-label pairing and a specific adversarial threat model. Spurious correlations, by contrast, encompass any statistical association between an artifact and a label, including naturally occurring ones (HTML patterns, metadata remnants, timestamp formats, financial markers, templated structures, etc.). Our work aims to characterize how LoRA behaves under these general conditions, not only in adversarial settings. In addition, the reviewer mentioned modifying the target label, but throughout our analysis we never modify or change the labels. We inject tokens into the samples and then allow the model to find correlations and shortcuts by relying on those injected tokens.
> > > >
> > > > We would also like to clarify a misunderstanding in the reviewer's comment: we never convert all examples to a single target label. The labels remain exactly as in the original dataset. What we inject is a token (a date token for the 100% examples), not a relabeled target. Moreover, this injected token is not a single fixed token: we use a distribution of many different date tokens, drawn from a wide range of values. This is fundamentally different from a backdoor-style single-trigger single-label poisoning setup. Because the targets are untouched, and the injected tokens vary across examples, the model is not trained on a "100% poisoned, single-label" distribution. Instead, it is exposed to a realistic form of systematic feature-label correlation, similar to naturally occurring metadata artifacts, which allows us to study LoRA's robustness properties rather than trivial label-collapse behavior. In addition, we want to further highlight that the figure displaying 100% is a worst case scenario meant to show extreme manipulation. This is precisely why our core experiments intentionally avoid using 100%. As shown in Sections 4.2-4.5, our main results focus on smaller proportions injected, where clean accuracy remains high and the model appears functional while still developing a shortcut that can be activated at inference. This is the realistic spurious-pattern regime in which **LoRA's non-monotonic robustness behavior emerges, and our contribution is to analyze that efficiency-robustness tradeoff in detail**.

---

### Author Response · Authors · 2025-11-30
**Overview for new AC**

We thank the new AC for their time. Below is an overview of key reviewer concerns and how our rebuttal addresses them.

Across the reviews, a point of confusion appears to be a perceived equivalence between our experimental methodology and classic backdoor poisoning attacks. We would like to clearly articulate why this interpretation does not reflect our setup, objectives, or contributions. First, although our work is related to the backdoor literature, **our study is not a backdoor attack method and does not follow the assumptions of that threat model**. Backdoor attacks require a fixed, intentionally designed trigger-target label mapping. In contrast, our experiments introduce spurious correlations, not fixed triggers, through a varied set of injected tokens. Crucially, we never modify any training examples. The dataset remains perfectly balanced, and the only modification is the presence of correlated artifacts, analogous to naturally occurring metadata or html often found in real-world data. **If the mere presence of a correlated artifact were sufficient to classify work as "equivalent to a backdoor attack," then virtually all empirical research on spurious correlations, shortcut learning, and dataset artifacts would fall into the backdoor category. These fields exist independently because the underlying phenomena, assumptions, and goals differ fundamentally from adversarial trigger-based attacks**.

Second, we would like to correct a key factual misunderstanding: we do not "poison 100% of the training examples and change them to a specific target label." No labels are ever changed. In the 100% injection setting referenced by reviewer uKwD, every example retains its original ground truth label, and the injected tokens vary across samples. Thus, the model is not exposed to a single-label poisoned distribution, nor is it trained on a trivial majority-class scenario. This is why, if the phenomenon were simple overfitting or class imbalance, the model's predictions at test time would remain roughly 50/50. Instead, when we insert new, unseen tokens at inference, predictions collapse to ~99% for the associated class, a behavior that cannot be explained by the reviewer's proposed mechanism.

Third, we acknowledge that presenting the 100% injection case early may have unintentionally suggested that such a setting is central to our claims. It's not. That is explicitly a worst-case that establishes an upper bound. The core of the paper, and all main scientific conclusions, is based on the realistic regime examined in Sections 4.2-4.5, where only 25-75% of examples carry spurious tokens, clean accuracy remains high, and the model appears well-behaved yet still contains shortcuts. This regime captures both realistic data-quality issues and moderate adversarial settings without utility degradation. These are the settings in which our **central contribution, the non-monotonic relationship between LoRA rank and robustness, emerges.**

Fourth, we agree with reviewers that our framing may have over-emphasized adversarial language, inadvertently reinforcing the backdoor interpretation. We will adjust the narrative to center the actual scientific purpose: a robustness analysis of LoRA's efficiency-robustness tradeoff under spurious correlations. **SSTI is a controlled experimental tool, not an attack proposal**. Our ablations across ranks, model sizes (including 24B), tokens, and duration reveal architectural robustness properties specific to LoRA that aren't captured in backdoor work, including: low ranks being more robust under light corruption but less robust under heavy corruption, higher ranks showing the opposite behavior, and robustness not improving with scale or more training. **These phenomena do not arise from classic backdoor dynamics and, to our knowledge, have not been documented previously.**

Finally, regarding theoretical explanation and sequence-level extensions: our choice of single-token perturbations was deliberate, allowing us to isolate LoRA's capacity-robustness behavior without conflating factors like compositional semantics. Single-token artifacts also mirror many real-world sources of spuriousness (HTML, dates, etc). Nevertheless, we agree that sequence-level effects are an important extension and will include preliminary results in the appendix while clarifying this methodological choice in the main text.

We hope these clarifications address the central concerns. The methodological misunderstandings regarding "label modification" and "single trigger insertion" were foundational to several critiques, and we believe that resolving them makes our contributions clearer: **this work provides the first systematic analysis of LoRA's robustness under spurious correlations and reveals a novel, non-monotonic efficiency-robustness tradeoff that is architectural in nature and independent of backdoor-style threat assumptions, having strong implications for anyone that finetunes a model using LoRA.**

---

### Meta-Review · Area_Chair_Avq1 · 2026-01-01

**Summary:**

This paper introduces SSTI (Seamless Spurious Token Injection) to demonstrate that LoRA can be manipulated by relying solely on a single token spuriously correlated with downstream labels. The evaluation is comprehensive, including the effect of LoRA hyperparameters and some potential defenses. However, the AC agrees with the reviewers' concerns regarding the novelty of the proposed SSTI framework, as it does not appear fundamentally distinct from the well-established poisoning-based backdoor attacks. Furthermore, the relationship between LoRA rank and SSTI effectiveness lacks theoretical grounding. While the study addresses an important topic in LoRA security, the attack setting could be more realistic, such as the poisoning rate. It is recommended that the paper include more real-world examples of "spurious tokens" to better illustrate the practical relevance and motivation of this work. It would also be interesting to discuss how the attack performs on the variants of LoRA.

**Reviewer Concerns:**

- Unrealistic poisoning rates.
- Novelty & relation to backdoor attacks.
- The observed relationship between LoRA rank and SSTI effectiveness lacks a theoretical explanation.

**Reviewer Scores:**

I believe the current scores are appropriate.

---

### Decision · Program_Chairs · 2026-01-26

Reject